# Depressive Symptoms and Control of Emotions among Polish Women with Polycystic Ovary Syndrome

**DOI:** 10.3390/ijerph192416871

**Published:** 2022-12-15

**Authors:** Karolina Pokora, Karolina Kowalczyk, Agnieszka Wikarek, Małgorzata Rodak, Karolina Pędrys, Mariusz Wójtowicz, Katarzyna Wyskida, Mariola Jonderko

**Affiliations:** 1Department of Endocrinological Gynecology, School of Medicine in Katowice, Medical University of Silesia, 40-752 Katowice, Poland; 2Pathophysiology Unit, Department of Pathophysiology, School of Medicine in Katowice, Medical University of Silesia, 40-752 Katowice, Poland; 3Department of Gynecological and Obstetrics Women’s and Child Health Center, Medical University of Silesia, 41-803 Zabrze, Poland; 4Health Promotion and Obesity Management Unit, Department of Pathophysiology, School of Medicine in Katowice, Medical University of Silesia, 40-752 Katowice, Poland; 5Department of Internal Medicine and Oncological Chemotherapy, School of Medicine in Katowice, Medical University of Silesia, 40-029 Katowice, Poland

**Keywords:** Polycystic ovary syndrome (PCOS), BMI, depression, Beck Depression Inventory (BDI), emotional control, Courtauld Emotional Control Scale (CECS)

## Abstract

Introduction: Polycystic ovary syndrome (PCOS) is a disorder that substantially affects women’s health. It is particularly diagnosed in young patients. Women with PCOS are burdened with excessive weight gain, overweight and obesity (74%) compared to a healthy female population. Excessive weight influences psychological state and emotional well-being, whereas in the meantime, psychological and behavioral dysfunction is increasingly being diagnosed among patients with PCOS. Aim: To assess depressive symptoms and emotional control among women with PCOS in relation to BMI. Methods: The study was conducted among 671 self-reported PCOS women. The standardized Beck Depression Inventory (BDI) was used to assess depressive disorders. Emotion control was assessed using the Courtauld Emotional Control Scale (CECS). Results: Moderate and severe depressive symptoms were more common in PCOS women with abnormal BMI compared to normal BMI subjects (*p* < 0.01). In total, 27.1% of obese women had moderate depression and 28.8% had severe depression. Among overweight women, 19.9% suffered from moderate and 25% from severe depressive symptoms. Underweight women also reported moderate (25.6%) and severe (33.3%) depressive signs. There were no statistically significant differences between the body weight of the women studied and the CECS scores. Conclusions: Depressive symptoms are more common in women with PCOS and abnormal BMI than in women with PCOS and proper BMI. The severity of depressive symptoms increases with BMI, but underweight women with PCOS are also at risk of depressive disorders. The level of suppression of negative emotions is independent of BMI in women with PCOS.

## 1. Introduction

Polycystic ovary syndrome (PCOS) is an endocrine disorder diagnosed especially among young women [1]. The prevalence of this condition is estimated at 8–13% of reproductive-age women [2]. However, it is estimated that 70% of patients are undiagnosed for PCOS [2]. PCOS diagnosis is based on modified Rotterdam criteria, whenever at least two of the following are required: oligo- or anovulation, clinical and/or biochemical hyperandrogenism and polycystic ovaries on adult ultrasound (number of follicles per ovary ≥20 and/or ovarian volume ≥10 mL) [2]. This condition is heterogeneous and has different phenotypes [3,4]. The etiology is not fully known. It has been hypothesized that PCOS is a complex polygenic disorder. Furthermore, its development is influenced by the environment, diet, and lifestyle [5]. PCOS is most often accompanied by significant metabolic comorbidities. The most frequently observed are insulin resistance, impaired glucose tolerance, diabetes mellitus, abdominal obesity-metabolic syndrome, or dyslipidemia [5]. Non-alcoholic fatty liver disease, coagulation disorders, or hypertension are also frequent abnormalities [6].

Women with PCOS have a 2–3 times greater risk of being overweight or obese than women without PCOS. Both genetic and environmental factors are involved in pathogenesis. This problem affects up to 80% of patients with PCOS, having increased both BMI and waist-to-hip ratio [7]. Studies have shown that the central distribution of body fat is most clinically significant and can have serious consequences, such as cardiovascular diseases [8]. Visceral fat is also increased in patients with PCOS [9] and normal BMI compared to women without PCOS, which has metabolic and endocrine consequences. Hyperandrogenism and obesity are mutually correlated—excessive body fat increases androgen levels, while hyperandrogenemia may contribute to abdominal obesity in patients with PCOS [7,8].

Patients with PCOS may have eating disorders, where anorexia nervosa, bulimia nervosa or binge eating disorder are primarily mentioned. Prevalence of disordered eating is increased in patients with PCOS compared to control [10]. It ranges from 12% to 36% [11,12], but the result may be even higher in this group of patients [10]. Eating disorders are psychiatric conditions, which can affect the patient’s psychological and physical state. At the same time, eating disorders correlate with body weight, which can also affect mental health [10]. Jeanes et al. reported the prevalence of binge eating and food cravings and their relation to obesity risk in women with PCOS. The results indicate that a high proportion of women with PCOS exhibit binge eating behaviors [13]. Larson et al. showed that women with PCOS achieved higher scores of uncontrolled eating or emotional eating compared to controls after adjustment for age and BMI [14]. It has been confirmed that anxiety increases the risk of disordered eating fivefold in women with PCOS [15]. These inappropriate eating habits may contribute to excess weight gain and abnormal body mass.

The above-mentioned aspects related to PCOS may affect the mental state and predispose to the mood and emotional disorders. PCOS patients are also at risk for various psychiatric disorders like depression, generalized anxiety disorder, personality disorders, social phobia, obsessive-compulsive disorder, attention deficit hyperactivity disorder (ADHD), eating disorders, schizophrenia, or bipolar affective disorder [16]. Previous studies suggested that women with PCOS are prone to depressive symptoms and depression, irrespectively from obesity [2]. Moreover, recent studies have shown that the prevalence of depressive and anxiety symptoms remains high with age [17]. The cohort study found that depression and anxiety scores significantly increased in both 31- and 46- year-old PCOS women compared with controls. These results suggest that symptoms of mood disorders are persistent in these women regardless of age [18]. The prevalence of depression in the PCOS population is higher than in the general population [3,19]. It affects between 14 and 67% of females with PCOS, whereas the prevalence in general population is less than 4–6% [20]. The reasons for the higher rate of depression in women with PCOS are not precisely defined, and various factors may be involved [3,19]. Psychological distress is possible in women with PCOS, caused mainly by PCOS symptoms related to physical appearance. Chronic stress and abnormalities of the hypothalamic–pituitary–adrenal (HPA) axis increase cortisol levels, which may contribute to the development of depressive disorders. Another risk factor may be vitamin D deficiency, common in this group of patients. It is also worth mentioning that there is a possible link between elevated inflammatory markers in depression and pro-inflammatory markers in PCOS [3]. Infertility, which is more frequent in this group of patients than in the general population, may also play a role [19].

Depression and overweight or obesity are global problems that are increasingly co-occurring. Both conditions represent a complex problem with many contributing factors [21]. Biological, psychological, immunological, endocrinological or molecular factors are considered linking them [22]. Obesity and depression influence each other, since long-term depression can lead to obesity and vice versa. Four longitudinal meta-analyses confirm the existence of this bidirectional relationship [22]. Patients with obesity have a 55% increased risk of developing depression, whereas patients with depression had a 58% increased risk of becoming obese [23]. Effect sizes are positive but less strong and not always significant for overweight (BMI 25–30 kg/m^2^) [22,24]. These findings have important clinical relevance: a patient with depression should have body weight monitored, and at the same time, a patient with obesity should have mental health checked.

Thus, the purpose of this study was to evaluate symptoms of depression and emotional control among women with PCOS in relation to BMI.

## 2. Materials and Methods

### 2.1. Patients

For the purposes of this study, we surveyed women from the beginning of November to the end of December 2020. All survey data were collected using an online format. In total, 784 internet surveys were collected among women with self-reported polycystic ovary syndrome. A total of 113 questionnaires were excluded due to taking antidepressants and psychotic drugs. Google forms have been distributed among several groups of women with polycystic ovary syndrome. Subjects were informed of the study characteristics and told that the data obtained from questionnaires would be entered into a database with an identifier code, ensuring patient anonymity. All participants provided their consent and did not receive any remuneration for completing the survey.

### 2.2. Methods

The study used an original, anonymous, 41-item self-report questionnaire concerning sociodemographic and clinical data. The questionnaire included questions on weight, height, taking up physical activity, menstrual cycle, and hormonal contraception and questions screening for PCOS. The survey is presented in the Appendix A. It contained closed, open-ended, and multiple-choice questions, including the answer “Other”, allowing for additional explanations.

In the context of PCOS, women were asked: “How many years ago were you diagnosed with PCOS by a doctor?”, “Has anyone in your family suffered from PCOS”, “Have you experienced any of the following symptoms? (e.g., acne, seborrhea, hirsutism, overweight/obesity)”, “Have you had elevated levels of androgens (testosterone, androstendione) in your laboratory results in the past or at present?”, “Was polycystic ovarian morphology present in the past or present ultrasound examination?”, “Do you suffer from any of the following ailments? (e.g., hypertension, diabetes, insulin resistance, impaired glucose tolerance, impaired fasting glucose, lipid disorders)”, “After being diagnosed with PCOS, have you attempted to comply with the following behaviours? (e.g., weight reduction, increased physical activity, diet with a limited caloric content)”, “Do you take medications for PCOS? If so, what medications do you take?”.

The standardized Beck Depression Inventory (BDI) was used to verify mood disorders in this group of patients. The questionnaire has 21 questions. Each question comprises four variants of answers, which are scored from 0 to 3. The total score of the questionnaire ranges from 0 to 63 points. Based on the sum of points obtained in Beck’s Inventory, one can differentiate the severity of symptoms of depression. Score ranges 0–11, 12–19, 20–25, and 26–63 were interpreted as representing no depression, moderate depression, and severe depression, respectively [25].

Emotional control was assessed based on the Courtauld Emotional Control Scale (CECS). It is a validated reference instrument to estimate the subjective level of the negative emotions expressed in difficult situations. The CECS comprises three subscales: Anger, Depression, and Anxiety, each with 7 statements. The Scale contains 21 statements. A maximum of 84 points can be scored. The sum of the points obtained from the three parts determines the total emotional control index. The CECS has no range, and therefore, the result obtained is compared between the study groups [26]. In our study, we analyzed the relationship between the BMI of self-reported women with polycystic ovary syndrome and the level of negative emotional control.

The examined patients were divided into four groups according to the value of the BMI. We asked the women about their body weight (kg) and height (m). Based on these values, the BMI was calculated as weight (kg)/height (m^2^). They were underweight (<18.5), normal weight (18.5–24.9), overweight (25–29.9), and obese (≥30).

### 2.3. Statistical Analysis

The basic parameters of descriptive statistics were calculated. The data are presented as mean (±standard deviation, SD), median (min–max), or percentages (*n* (%)), with percentages rounded to one decimal place. The Shapiro–Wilk distribution normality test was carried out. To assess the significance of differences between the variables, the Student’s t-test for independent samples and the non-parametric U–Mann–Whitney test were used. The Rank Spearman test was used to determine the correlation coefficient between the variables. The STATISTICA 13 PL software (TIBCO Statistica, v. 13.3.0, TIBCO Software Inc., Palo Alto, CA, USA) was used. *p*-values are only measured as descriptive measures and should be interpreted with caution. A *p* value of less than 0.05 was considered significant.

## 3. Results

A total of 671 questionnaires were collected among women diagnosed with PCOS. The mean age of the study group was 26.6 ± 4.8 years, and 77.2% of the respondents lived in urban areas; 67.4% declared physical activity lasting at least 30 min per day. The level of physical activity depending on BMI was shown in Figure 1. These activities did not include physical work, shopping, or housework.

More than half (52.2%) had an irregular menstrual cycle; 20.9% had children, and 26.4% used hormonal contraceptives, while 5.81% of the respondents were diagnosed with underweight; 45.6% had normal weight; 23.2% were overweight, and 25.3% were obese. Detailed characteristics of the group are presented in Table 1.

**Table 1 ijerph-19-16871-t001:** Characteristics of group.

Variable	Study Group (*n* = 671)	Underweight (*n* = 39)	Normal Weight (*n* = 306)	Overweight (*n* = 156)	Obesity (*n* = 170)	*p* Value
Age (years, avg. ± SD)	26.6 ± 4.8	25.1 ± 4.0	25.8 ± 4.4	26.9 ± 5.1	28.2 ± 4.9	<0.001
BMI (kg/m^2^), avg. ± SD	26.3 ± 6.5	17.6 ± 0.7	21.9 ± 1.8	27.1 ± 1.4	35.6 ± 4.7	<0.001
Physical activity *						
Yes (*n*/%)	452 (67.4)	15 (38.5)	213 (69.6)	110 (71.5)	114 (67.1)	<0.001
Menstrual cycle regularity **						
No (*n*/%)	350 (52.2)	23 (59.0)	141 (46.1)	80 (51.3)	106 (62.4)	NS
Having children						
Yes (*n*/%)	140 (20.9)	6 (15.4)	44 (14.4)	45 (28.9)	45 (26.5)	<0.001
Time since PCOS diagnosis						
<1 year (*n*/%)	167 (24.9)	19 (48.7)	89 (29.1)	35 (22.4)	24 (14.1)	0.02
1–5 years (*n*/%)	276 (41.1)	14 (35.0)	133 (43.4)	60 (38.5)	69 (40.6)	NS
>5 years (*n*/%)	228 (34.0)	6 (15.3)	84 (27.5)	61 (39.1)	77 (45.3)	<0.01
Hormonal contraception						
Yes (*n*/%)	177 (26.4)	13 (33.3)	85 (27.8)	52 (33.3)	27 (15.9)	NS

The patients were classified into different weight groups depending on the value of the BMI index: <18.5—underweight, 18.5–24.9—normal weight, 25–29.9—overweight, ≥30—obesity. * Physical activity for at least 30 min per day, which is not related to performing daily activities like work, shopping, and housework. ** The duration of the menstrual cycle is measured from the first day of bleeding to the first day of the next bleeding, ranging from 21–35 days, with an average of about 28 days. NS—no statistically significant.

**Figure 1 ijerph-19-16871-f001:**
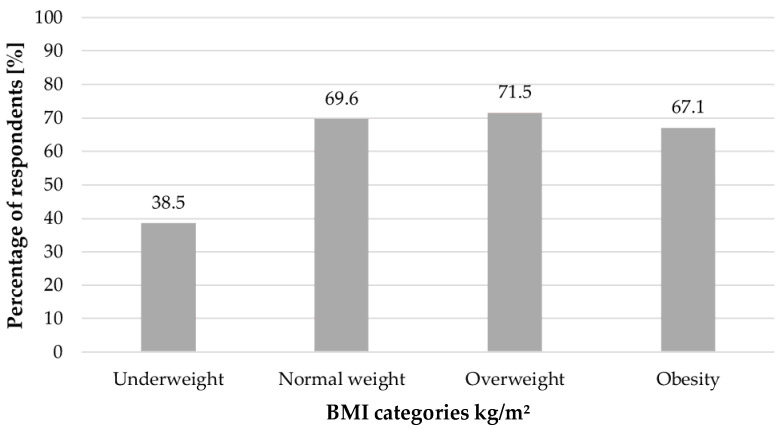
The level of physical activity depending on BMI (*p* < 0.001).

Of the study group, 28.3% presented mild depressive symptoms, 21.5% moderate, and 23.2% severe ones. Among obese, overweight, and underweight women moderate and severe depressive symptoms were found more frequently compared to normal weight subjects. Moderate depressive symptoms were presented by 27.1% of obese women, 19.9% of overweight patients and 25.6% of underweight women with PCOS. The symptoms of severe depression were presented by 28.8% obese patients, 25% overweight women, and 33.3% underweight females. The relationship between the moderate degree of depressive disorders according to BDI and BMI was statistically significant (*p* < 0.01) as well as for the severe degree of depressive disorders (*p* < 0.001). The results are summarized in Table 2.

As it can be observed (Figure 2 and Figure 3), moderate and high depression index according to the BMI category follows the J curve, which indicates that both too low body weight and excess body weight increase the risk of depressive disorders. However, the intensification of depressive symptoms increases with increasing BMI among self-reported PCOS patients.

The average result of the total emotional control index was 53 in the study group. Patients with normal BMI achieved 53.4 points, underweight women 52.5, overweight patients 52.6, and obese women 52.8. These data are presented in Figure 4. The mean level of the Anger subscale was 15.7, of the Depression subscale 19.1 and of the Anxiety subscale 18.2. The mean scores in all groups of women were comparable. There were no significant statistical differences between BMI and the results of the CECS scale (*p* > 0.05). Only the level of Anger suppression was on the verge of statistical significance depending on BMI (*p* = 0.08) (Table 2).

## 4. Discussion

The problem of depressive disorders in this group of patients is notable. A review of 46 studies and four meta-analyses conducted separately found that women with PCOS, compared to healthy women, were significantly more prone to psychiatric disorders, such as depression [27]. The prevalence of depression in women with PCOS is estimated in 36.6% [2]. However, our study confirmed the presence of depressive symptoms in 73% of respondents. Research demonstrating a correlation between mood disorders and PCOS has mainly been conducted in adults [2]. Women with PCOS experienced major depressive disorder more frequently compared to the control group and were more often hospitalized for this reason. This was revealed in a retrospective cohort study, after adjusting for various confounding variables, including BMI. Women in the study group with a BMI of 25 had a higher risk of hospital admission for depression than women with a BMI below 18.5 [19]. However, our research emphasizes that both obesity, overweight and weight deficiency favor the development of moderate and severe depressive symptoms in PCOS. Data collected at the 2006 Behavioral Risk Factor Surveillance System demonstrated that depression and anxiety disorders are significantly more common not only in overweight or obese patients but also in underweight patients (BMI < 18.5 kg/m^2^) [28]. Most studies assessing the impact of BMI on developing mental disorders consider obesity, but often neglect the effect of underweight. Analysis of cross-sectional studies revealed higher rates of anxiety and depression diagnosis in underweight patients compared to controls of normal-weight patients [21]. At the molecular level, there is a strong correlation between decreased level of leptin and depressive symptoms in underweight individuals [29]. These results are consistent with our study.

PCOS patients are at high risk of developing depression as well as excess body weight [24]. According to our research, the prevalence of moderate and severe depressive symptoms increases in overweight and obese PCOS women. However, obese patients are at greater risk than overweight women. Current research suggests that obesity predisposes to develop depression both in the general population and among women with PCOS. Naumova et al. showed that the severity of depression was statistically significantly associated with obesity, hirsutism, and testosterone concentration (*p* < 0.01). The results obtained among infertile patients with PCOS indicated that 48.6% of them had depressive symptoms [30]. BMI had a strong association with depressive symptoms [2]. Stapińska-Syniec et al. studied a group of 250 women with PCOS. Of these, 52% had depressive symptoms, and their BMI was significantly higher than the rest of the study group as well) (*p* < 0.001) [20]. According to Joshi R. et al., 51.4% women among 105 diagnosed PCOS patients, aged between 18 and 50 years, developed depressive symptoms. This data justifies why women with this gynecological disease should have access to psychiatric care [31]. On the other hand, Besenek and Gurlek did not confirm a relationship between body weight and mental state among adolescent PCOS patients. The result was obtained in PCOS adolescent, categorized into three subgroups according to BMI, compared to the control group. However, there were small study subgroups [32]. Another study revealed increased prevalence of moderate and severe anxiety and depressive symptoms regardless of BMI [33]. As the results of numerous studies are inconclusive, it is worth conducting further studies in this area. To sum up, BMI in most studies, as well as in ours, was associated with an increased risk of depressive disorders (*p* < 0.001).

Besides excess body weight, symptoms of PCOS may play a role in the development of depression. The presence of them could be considered a confounding factor in our research. Especially hirsutism, which is a clinical manifestation of hyperandrogenism, may be associated with an increased occurrence of depression [33]. It has a significant impact on the patient’s physical appearance, reduced mood, or embarrassment. On the other hand, acne, which is also common in PCOS women, does not increase their risk of mood disorders [34,35]. Another aspect is insulin resistance, which correlates with high androgen concentrations. It has been suggested that their increased levels play a role in promoting mood disorders [36]. The duration of exposure to hyperandrogenism corresponds to a higher chance of developing anxiety or depression [3].

In our research, we also considered the fertility of patients. Reduced reproductive capacity or lack thereof because of PCOS results in increased anxiety or other emotional disturbances. The fertility significantly influences the self-perception of the patient. However, the severity of the impact of infertility on a woman’s life depends on socio-cultural factors and varies among populations [35]. Diagnosed infertility among women with PCOS is a factor that reduces the psychological well-being of these women, compounding anxiety or depressive symptoms [37]. In our study group, 20.9% of the women had children. However, the study was conducted in the Polish population, which is characterized by a tendency to postpone maternity. According to the data presented by Eurostat, the mean age of Polish women at birth of the first child was over 28 years in 2020 [38]. Perhaps the surveyed women have not yet started their maternity plans, which had an impact on our results.

The next factor affecting mental health is physical activity. Exercise improves the mental well-being of women with PCOS, but the optimal dose of physical activity that will guarantee the expected results is not established. PCOS guidelines suggest at least 150 min of physical activity per week [39]. Cross-sectional studies also indicate that inactive women with PCOS were more likely to suffer from mild depression [40]. The recommended first line PCOS therapy is a lifestyle change through implementing regular physical activity and a balanced diet for improving overall health and quality of life [40]. PCOS affects quality of life and can exacerbate anxiety and depression due to symptoms or the diagnosis of a chronic disease [2]. The benefits of making these changes include the improvement of depressive disorders in overweight and obese women with PCOS [41].

The presence of low-grade chronic inflammation may play a significant role in the pathophysiology of depression [10]. People with depression are characterized by increased levels of pro-inflammatory markers, such as interleukin (IL) IL-6, IL-1, tumor necrosis factor alpha (TNF-α) and C-reactive protein (CRP) [42]. Although physical activity increases the secretion of IL-6 and should promote pro-inflammatory effects, secretion of anti-inflammatory antagonists of the IL-10 and IL-1 receptor (IL-1ra) during exercise inhibits the production of inflammatory markers [43]. In addition, the release of adrenaline and cortisol during physical activity increases the secretion of IL-10, which reduces the concentration of TNF-α [44]. This means that regular exercise can extend the anti-inflammatory period, which may play an important role in the pathomechanism of depression. However, our study achieved no significant correlation between the frequency of undertaking physical activity and the result obtained in the BDI. Perhaps, the respondents made an effort inadequate to their needs, due to the impossibility of establishing the optimal and patient-specific dose of physical activity. In addition, a non-standardized tool was used to assess physical activity among the respondents.

Oral contraceptive pills (OCPs), which affect mental and psychological states, are frequently used in the treatment of PCOS [45]. They can reduce menstrual irregularity, hyperandrogenism, and hirsutism and lead to improved quality of life. They influence the clinical features of this syndrome, as well as depressive symptoms, anxiety, poor body image, and low self-esteem [2]. Our research did not show any correlation between the use of OCPs and symptoms of depression. However, a previous study suggested that oral contraceptive pills are not indifferent to mental health [46]. Both positive and negative effects on psychiatric disorders have been reported among women using OCPs [47]. There are reports of both improvement and no improvement after the use of these drugs in depressive symptoms among women with PCOS [48].

In the second part of our study, we focused on emotional control in self-reported PCOS women related to BMI. Current research does not provide information on the level of negative emotional suppression among women with PCOS. However, awareness of a disease generates various emotions, especially negative feelings. People affected by the illness may find it difficult to manage their emotions [49]. Women diagnosed with early breast cancer reveal a higher degree of emotional suppression than healthy females do [19]. In fact, high emotional control generates a negative impact on the mental health of people with cancer [50]. The average values of the total CECS obtained in the study were higher than the norms for the women presented by Juczyński (53.0 vs. 49.97) [51]. The same is true for the Depression (19.1 vs. 16.88) and Anxiety (18.2 vs. 17.08) subscales. The Anger score was of comparable value (15.7 vs. 16.01). Our mean result was higher than in a group of women during menopause (51.83) and dialysis patients (51.23), lower than a group of diabetics (55.77) and comparable to men after myocardial infarction (53.03) [26]. Therefore, women with PCOS suppress negative emotions more strongly than healthy women do. However, the level of negative emotions suppression in women with PCOS is independent of BMI. Only the level of Anger suppression was on the verge of statistical significance according to BMI (*p* = 0.08). Unexpressed negative emotions, which repeat and last for a long time, may underlie neurotic disorders and psychosomatic diseases [26]. Considering that PCOS patients are predisposed to various psychiatric disorders, suppression of negative emotions may be the first step in the development of these diseases. According to Lingyan Li and al., emotional suppression plays a role in exacerbating depressive symptoms among women newly diagnosed with breast cancer. These researchers emphasize that the suppression of anger may play an important role in predicting the development of depression [16].

Furthermore, direct expression of feelings plays an important role in managing the disease. Low emotional control is conducive to increasing the level of acceptance of the disease. It is a protective factor for mental health and can improve quality of life [50]. Women with PCOS are at risk of a variety of mental diseases [49], and therefore, it is important to influence modifiable risk factors such as body weight and its effect on emotional control in women with PCOS. However, our study reported that BMI has no effect on the expression of emotions. On the other hand, female patients surgically treated for gynecological reasons presented a higher level of suppressed negative emotions in the perioperative period than before the surgery, regardless of the category of medical procedure [16]. The values were varied depending on age and level of education. The CECS result was higher in patients over 40 years of age compared to younger women. This result was consistent with the standard data presented by Juczyński, which shows that women under 40 present a lower level of emotional suppression than women over 40 [26,51]. In contrast, our study group included women of reproductive age. Patients constituted a homogeneous group in terms of age. The mean age of responders was 26.6 ± 4.8 years. Therefore, the influence of age was not a significant aspect in our research.

The COVID-19 pandemic was the “temporal background” of our research. Thus, it may have been a confounding factor in our results. According to current research, fear of getting ill, as well as the SARS-CoV-2 infection itself, can have a negative effect on mental health. COVID-19 can generate inflammation in the brain [52].

Women with PCOS are vulnerable to a deterioration in their mental well-being [53]. PCOS patients are substantially exposed to an increase in depressive disorders, stress levels, sleep quality disturbances, or difficulty maintaining weight during the COVID-19 pandemic [54]. A mental state may be associated with lots of restrictions as difficult access to medical services, including infertility treatment, limited interpersonal contacts, or socioeconomic difficulties because of loss of employment during the lockdown [53,55].

## 5. Strengths and Limitations

To our knowledge, this is one of the few studies directly comparing the presence of depressive symptoms and emotional control among patients diagnosed with PCOS, showing that the severity of depressive symptoms increases with BMI, but that underweight women with PCOS are also at risk for depressive disorders. We were able to identify several modifiable factors that may influence BMI values in the above study group. From a clinical point of view, due to the high prevalence of emotional disorders among these patients, measures should be taken to reduce their severity in order to improve mental health. Appropriate psychological support may be helpful. It can also be suggested that clinicians use appropriate scales, such as the standardized Beck Depression Inventory (BDI), to identify the disorder-prone population. The above data were collected from questionnaires completed online, due to the safety of female study participants during the COVID-19 pandemic. The study has also some limitations. The study was conducted through an online questionnaire filled out by women with PCOS. We confirmed their diagnosis based on a number of symptoms that were included in our questionnaire. It was primarily aimed at assessing the patient’s current severity of depressive symptoms and emotional control through the questionnaire. We only have data on the severity of the above symptoms, and these symptoms may have varied, depending on the duration of the disease. In addition, the study was conducted during the COVID-19 pandemic, which intensified the prevention of depressive disorders, which may also affect the results we present. It is also noteworthy that the presence of various restrictions put in place during the COVID-19 pandemic, which were changing rapidly, may have affected the mental health of women with PCOS. However, we believe that none of these factors significantly affected the results of the study.

## 6. Conclusions

Depressive symptoms are more frequent in women with PCOS and abnormal BMI than in women with PCOS and proper BMI. With the rise of BMI, the degree of depressive symptoms increases. However, PCOS women who are underweight should be treated as a separate risk group for depressive disorders. All things considered, we recommend screening for these conditions in every PCOS patient using simple validated tools at diagnosis. Female patients with self-reported PCOS have a higher result emotional control compared to the group of standardized women. However, the level of suppression of negative emotions is independent of BMI in women with PCOS. These issues should be a collaborative effort between the clinician and the psychologist. The psychological state of patients can translate into efficacy of therapy or physical impairment, and thus should be addressed on an equal level as organic disease affecting PCOS patients.

## Figures and Tables

**Figure 2 ijerph-19-16871-f002:**
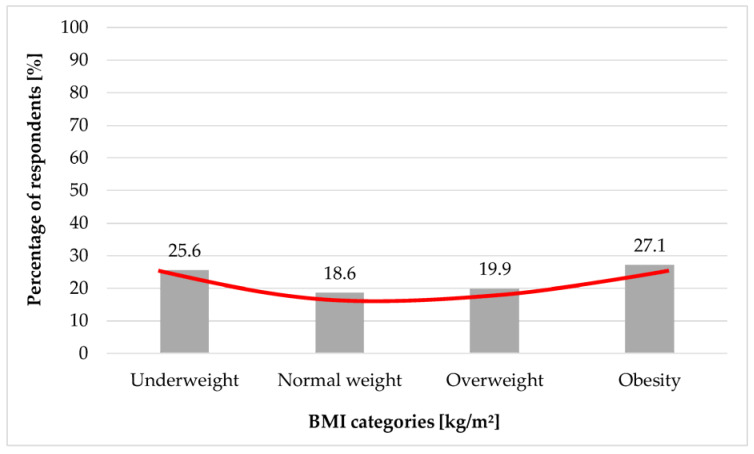
Percentage of individuals presenting a moderate depression index according to Beck’s scale (*p* < 0.01). The red line represents the J-curve.

**Figure 3 ijerph-19-16871-f003:**
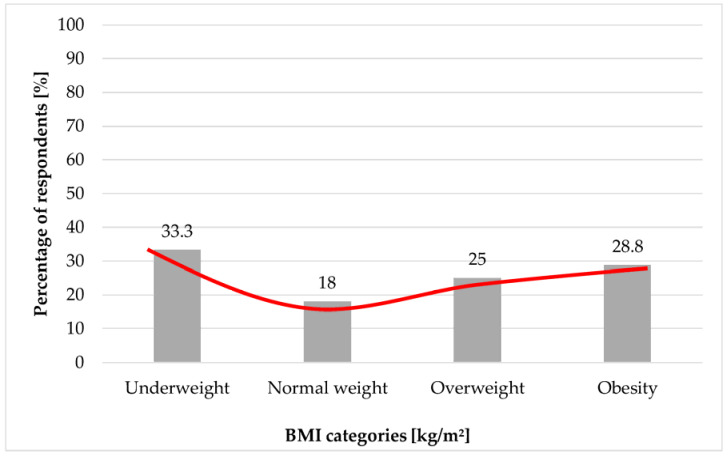
Percentage of individuals presenting a high depression index according to Beck’s scale (*p* < 0.001). The red line represents the J-curve.

**Figure 4 ijerph-19-16871-f004:**
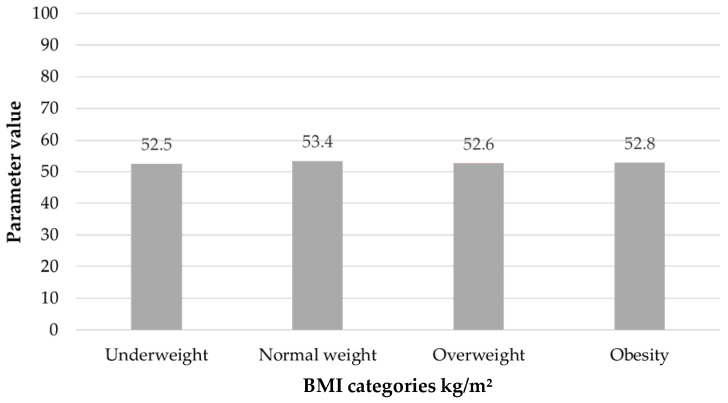
CECS- Total score depending on BMI (*p* = NS).

**Table 2 ijerph-19-16871-t002:** Depression levels and emotional control according to BMI.

	Study Group (*n* = 671)	Underweight (*n* = 39)	Normal Weight (*n* = 306)	Overweight (*n* = 156)	Obesity (*n* = 170)	*p* Value
Beck’s Depression Inventory						
Mean result (*n* ± SD)	18.3 ± 9.9	20.6 ± 10.8	16.4 ± 9.5	18.7 ± 10.3	20.9 ± 9.5	<0.001
Levels of Depression *						
No depression (*n*/%)	181 (27.0)	9 (23.1)	104 (34.0)	40 (25.6)	28 (16.5)	NS
Mild depression (*n*/%)	190 (28.3)	7 (18.0)	90 (29.4)	46 (29.5)	47 (27.6)	NS
Moderate depression (*n*/%)	144 (21.5)	10 (25.6)	57 (18.6)	31 (19.9)	46 (27.1)	<0.01
Severe depression (*n*/%)	156 (23.2)	13 (33.3)	55 (18.0)	39 (25.0)	49 (28.8)	<0.001
CECS						
Total score	53.0	52.5	53.4	52.6	52.8	NS
Anger	15.7	15.3	16.0	15.2	15.8	NS
Depression	19.1	19.4	19.1	19.3	18.9	NS
Anxiety	18.2	17.8	18.3	18.1	18.1	NS

The patients were classified into different weight groups depending on the value of the BMI index: <18.5—underweight, 18.5–24.9—normal weight, 25–29.9–overweight, ≥30—obesity. * Based on the sum of points obtained in Beck’s scale, the degree of depression can be differentiated: 0–11—No depression (lack of depression), 12–19—Mild depression (first degree), 20–25—Moderate depression (second degree), 26–63—Severe depression (third degree). NS—no statistically significant.

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
