# Peer review of "Depressive Symptoms and Control of Emotions among Polish Women with Polycystic Ovary Syndrome"

_ijerph, 2022, doi:10.3390/ijerph192416871_

Round 1

Reviewer 1 Report

Statistical analysis does not reveal any association between COVID-19 and its possible effects on mental health of the subjects. None of the questionaries include any detail on COVID-19. It's hard to elaborate how the authors came up with this association. Either this methodology needs to be corrected or materials and methods section needs to be revised.

Reviewer 2 Report

A study survey should be added at least like a supplementary file.

Moreover paper would benefit if at least some of the results (directly from the survey) were presented as figures. Now author only focuses on depression and BMI. On the other hand, the original survey provides an additional 41 questions, which could enhance the quality of the paper if adequately described.

Unfortunately, the COVID-19 factor in this study is irrelevant because the authors did not present any retrospective control (for any presented data ) group to which the reader could compare the results.

Line 170-178 – please provide references for CECS

Figures 1, 2 – the quality of the figures is bad. There is a need for higher resolution.

The paper lack a study limitation section.
